# Differences in the growth rate and immune strategies of farmed and wild mallard populations

Jana Svobodová[1]*, Hana Pinkasová[2], Pavel Hyršl[3], Monika Dvořáčková[1], Lukáš Zita[4], Jakub Kreisinger[2]

**1** Faculty of Environmental Sciences, Department of Ecology, Czech University of Life Sciences Prague, Prague, Czech Republic, **2** Faculty of Science, Department of Zoology, Charles University, Prague, Czech Republic, **3** Faculty of Science, Department of Experimental Biology, Masaryk University, Brno, Czech Republic, **4** Department of Animal Science, Faculty of Agrobiology, Food and Natural Resources, Czech University of Life Sciences, Prague, Czech Republic

☯ These authors contributed equally to this work.

* svobodovajana@fzp.czu.cz

**Data Availability Statement:** All relevant data are within the manuscript and its Supporting Information files.

## Abstract

Individuals reared in captivity are exposed to distinct selection pressures and evolutionary processes causing genetic and phenotypic divergence from wild populations. Consequently, restocking with farmed individuals may represent a considerable risk for the fitness of free-living populations. Supportive breeding on a massive scale has been established in many European countries to increase hunting opportunities for the most common duck species, the mallard (*Anas platyrhynchos*). It has previously been shown that mallards from breeding facilities differ genetically from wild populations and there is some indication of morphological differences. Using a common-garden experiment, we tested for differences in growth parameters between free-living populations and individuals from breeding facilities during the first 20 days of post-hatching development, a critical phase for survival in free-living populations. In addition, we compared their immune function by assessing two haematological parameters, H/L ratio and immature erythrocyte frequency, and plasma complement activity. Our data show that farmed ducklings exhibit larger morphological parameters, a higher growth rates, and higher complement activity. In haematological parameters, we observed high dynamic changes in duckling ontogeny in relation to their morphological parameters. In conclusion, our data demonstrate pronounced phenotype divergence between farmed and wild mallard populations that can be genetically determined. We argue that this divergence can directly or indirectly affect fitness of farmed individuals introduced to the breeding population as well as fitness of farmed x wild hybrids.

## Introduction

The restocking of free-living populations with farmed individuals (also called hand-reared, captive, captive-bred, captive-reared individuals) is one of the most controversial of

**Funding:** This study was supported by the Czech Science Foundation (GAČR Project No. 14-16596P) and the Grant Agency of the Faculty of Environmental Sciences at the Czech University of Life Sciences, Prague (Project No. IGA 20144268 and IGA 20154241). The funders had no role in study design, data collection and analysis, decision to publish, or preparation of the manuscript.

**Competing interests:** The authors have declared that no competing interests exist.

anthropogenic interventions in natural populations [1,2]. While farmed individuals are mainly released to increase numbers and the genetic diversity of endangered species [3–5], restocking is also commonly used to maintain or increase the abundance of free-living animals of economic concern, such as Atlantic salmon (*Salmo salar*) [6], red-legged partridge (*Alectoris rufa*) [7] or common pheasant (*Phasianus colchicus*) [8].

High rates of genetic drift, inbreeding [9] and altered or relaxed selection frequently occur in populations that have been held in captivity for many generations [10,11]. This can lead to a decrease in genetic diversity and phenotype divergence compared with free-living populations of the same species. Waterfowl species raised in captivity, for example, exhibit reduced brain volume [12] and shorter and lighter small intestines and caeca, which can reduce their ability to digest a natural diet [13,14]. As such, releasing of farmed individuals, and subsequent hybridisation with their free-living counterparts, can disturb the genetic integrity of natural populations, causing gradual phenotypic shifts, thereby decreasing their overall fitness [1,15,16].

From the 1970s on, the mallard (*Anas platyrhynchos*), one of the most widespread duck species, has been subject to massive restocking to increase hunting opportunities [1,17]. In Europe alone, almost 3 million farmed individuals are released every year [18], which is of comparable order of magnitude to European-wide breeding population of mallard (ca. 2.9–4.6 million breeding pairs) [19]. Farmed mallards exhibit clear genetic divergence and decreased genetic variation compared to the native population [20]. Despite long-term massive restocking, native genotypes are still widely preserved in today's natural populations [21,22], suggesting low survival rates for released individuals [14,23,24]. Nevertheless, presence of admixed individuals in natural populations indicates ongoing introgression of the farmed genotype into the wild gene pool [20–22]. Thus, it is tempting to speculate that long-term massive restocking could result in adverse consequences, including threats to genetic diversity and gradual phenotype shifts within natural mallard populations.

To date, however, information on phenotypic differences between wild and farmed mallards is rather scarce. Most previous studies have only compared individuals that were raised in the two distinct environments (e.g. [14,23,25,26]) and, as such, such studies fail to disentangle genetic vs. environmental effects on any phenotypic differences observed. Other studies have focused on historical phenotypic shifts that took place in a free-living population following the establishment of intensive mallard restocking (e.g. [27–29]). However, the phenotypic changes observed in such cases will not necessarily have been caused by restocking alone, but also other factors, such as habitat and climatic change, may be of comparable importance [30]. To isolate the effect of genotype from other confounding factors on phenotypic variation under wild and captive conditions, experiments conducted on individuals reared under the same environmental conditions are indispensable. To our knowledge, however, no study has yet been undertaken using this experimental design.

The aim of our contribution was to search for phenotypic variation between wild and farmed mallards during the early phase of post-hatching development (up to age of 20 days) under controlled conditions. Our focus on the early post-hatching phase stems from the high mortality rates observed in natural populations [31,32], which leads to strong selection on phenotypic traits. Furthermore, there are almost no data comparing the phenotypes of wild and farmed mallards during this life phase [33]. By using a common-garden experimental framework, where eggs are incubated and ducklings reared under the same conditions, we were able to suppress the effect of environmental variation, and thus provide more direct insights into genetic-dependent differentiation in phenotype between wild and farmed mallard populations. In addition to studying morphological traits, we also investigated variation in two haematological parameters, the stress-linked heterophil vs. lymphocyte ratio (hereafter H/L) [34] and the proportion of immature erythrocytes, the latter being positively linked to haematopoiesis rate

[35]. Moreover, we measured antibacterial activity of the plasma complement as an indicator of baseline innate immunity. The complement cascade links innate immunity and provides the first line of defence against the spread of infection [36]. Captive environment, typified by non-limited access to food and a relatively low effect from environmental stressors such as pathogens and predators, usually selects for genotypes allocating more resources to grow compared to other self-maintenance systems [1,37]. Consequently, we predicted that farmed populations will display higher rate of growth and haematopoiesis compared with wild mallard populations. We also predicted that immune function will be down-regulated in captive-reared individuals as these are typically exposed to lower pathogen variability [38–40]. On the other hand, farmed mallards may be expected to show superior functioning and greater investment in the immune system as a consequence of regular access to energy resources and low energy expenditure, which may relax trade-offs between immune function and body growth [41]. As the H/L ratio acts as an indicator of stress, we also expect a higher H/L ratio in wild ducklings due to a higher susceptibility to stress in captivity.

## Material and methods

### Population samples

The experiment was conducted in the Czech Republic where at least 170 000–200 000 farmed mallards are released every year (Czech Ministry of Agriculture 2009–2015) whereas the breeding population is estimated at 25 000–45 000 pairs [42]. The wild population was represented by eggs (n = 37) collected from free-living mallard populations at four localities (Fig 1, not more than two eggs collected per nest). Based on the results of our previous study [22], it is known that farmed genotypes are almost absent (<5%) at these localities. To eliminate the effect of incubation on postnatal development, we only selected eggs from non-incubated wild

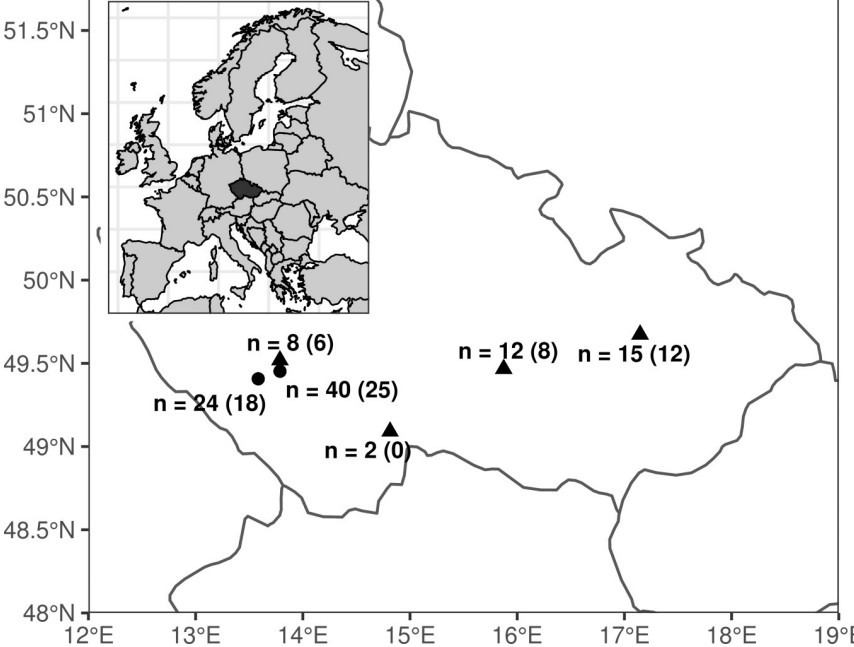

**Fig 1. The geographical distribution of the localities (Czech Republic) where mallard eggs were sampled is indicated by triangles (free-living populations) or circles (duck game-farms).** Sample sizes (n) for each location is shown together with numbers of hatched individuals, in parentheses.

clutches or from those that were in the very early stages of incubation (< four days of incubation according to Weller [43]. Non-incubated eggs from farmed populations (n = 64) were obtained from two duck game-farms belonging to hunting associations (Stráže Lnáře a.s. and Klatovského rybářství a.s.; Fig 1). As mallards lay one egg per day [44,45], newly laid eggs of farmed mallards were randomly selected over a single day, ensuring that they came from different females.

## Common-garden experiment

The common-garden experiment was conducted from the beginning of May to the beginning of July 2014 at the experimental facilities of the Czech University of Life Sciences. The eggs were cleaned and individually marked and their length (L) and width (W) measured with digital callipers (0.01 mm accuracy; Kinex, Prague, Czech Republic). Egg volume, a proxy of maternal energetic investment [46], was calculated as $V_{egg} = C_V \times L \times W$, where $C_V$ is a volume constant assessed according to mallard empirical data as $C_V = 0.515$ [47].

All eggs were then placed into an incubator with automatic egg turning (OvaEasy 190 Advance, Brinsea Products Inc., Buckingham, UK) and incubated at 37.5˚C and 50% humidity. As recommended by the manufacturer, incubating temperature was decreased to 37.3˚C, humidity increased to 80% and egg turning turned off on the 25th day of incubation. To identify the hatched ducklings, cracked eggs were placed in a separate net sack with an appropriate identity code [48].

All ducklings were marked with coloured rings at the age of one day and placed into indoor cages (88 × 48 × 45 cm [L/W/H]) with eight individuals of the same origin (wild vs. farmed) in each cage. Birds were provided with young duckling pellets (duckling feed KCH-1, VELAS a. s., Czech Republic) ad libitum and had permanent access to water. Photoperiod was set at 14:10 (light:dark) and room temperature at 22 ± 1 ˚C.

All ducklings were regularly measured throughout the experiment, with the same morphological parameters being recorded on the first day and at four-day intervals subsequently until the 20th day (i.e. six measurements in total; on the 1st, 4th, 8th, 12th, 16th and 20th day). On each occasion, body weight (0.1 g accuracy; PCB 6000–0, KERN & SOHN GmbH, Germany), length of right and left tarsus (from which mean tarsus length was calculated) and bill length (from the tip to the feathering) and width (the maximal width in distal part) were recorded. All morphological parameters were measured using a digital calliper (0.01 mm accuracy; Kinex 6040.2, Czech Republic) by the same person (HP). As leukocytes profile can change rapidly by short-term stress (e.g. by handling, [49]) a drop of blood was taken from the jugular vein on the 3rd, 9th and 15th day (i.e. one day before morphometric measurements) and used to prepare a blood smear for haematological analysis. Finally, 200 μl of blood (syringe 0.5 ml, needle 0.30 mm x 8.0 mm BD Micro-Fine) was taken from the 19-day-old ducklings and immediately centrifuged to obtain plasma samples. The separated plasma samples were then stored at -80˚C until analysis. The ducklings were provided to various breeders after completion of the experiment. The research was approved by the Ethical Committee of the University of Life Sciences in Prague and of the Central Commission for Animal Welfare at the Ministry of Education, Youth and Sports of the Czech Republic approved this research with animals (No. MSMT-31220/2014-5).

## Haematological assays

Differential leukocyte counts and frequency of immature erythrocytes were analysed from blood smears stained with Modified Wright-Giemsa Stain (product no. WG128, Sigma-Aldrich) and scanned using an Olympus CX-41 microscope (Olympus, Japan) under 1000×

magnification. The proportion of lymphocytes and heterophils was calculated from a sample of 120–160 leukocytes per smear (see [50]). Repeatability of the estimate was $r_{lymphocytes}$ = 0.93, $r_{heterophils}$ = 0.90, n = 10, p < 0.001; [51]).

The differential count of immature erythrocytes was estimated from 5–10 randomly chosen monolayer fields photographed at 100× objective magnification (ca. 800–1500 cells). Immature erythrocytes were manually counted from the photographs using ImageJ software 1.48 [52]. Repeatability of the measurement was assessed as r = 0.97 (n = 15, p < 0.001).

## Complement activity

Total complement activity was measured using the bioluminescence-based method described in Svobodová *et al*. [53]. The repeatability of the measurement was high (r = 0.88, n = 10, p < 0.001). As a sufficient plasma volume was not collected for all individuals, complement activity was measured for 38 farmed and 23 wild ducklings only.

## Sexing technique

Sex of ducklings was determined according to the presence of PCR products of CHD1-W and CHD1-Z using the primers P2 and P8 [54]. Methods for DNA extraction and PCR reactions are described in Poláková *et al*. [55]. To reliably genotype males, PCR samples with one band were repeated twice.

## Statistical analysis

Growth curves were assessed for body mass and tarsus length (i.e. average value for left and right tarsus), while changes in structural body size were analysed using relative tarsus length (i.e. tarsus length divided by actual body mass) as a response variable. As variation in bill morphology has previously been proposed to change under captive condition [27,29], we also focused on growth curves for relative bill length (i.e. bill length divided by actual tarsus length) and relative bill width (i.e. bill width divided by actual tarsus length). Growth models for each morphometric parameter were fitted using mixed models with Gaussian error distribution, where age-dependent variation in the morphometric trait was modelled via third order polynomials [56]. All morphometric traits except relative bill length were $\log_{10}$ scaled prior to analysis to stabilise variation of residuals. To test whether a given trait varied between farmed and wild individuals or between males and females, the effect of origin, sex and their two-way interactions with age (including all polynomial terms) were included as explanatory variables. Individual-specific and cage-specific variation in growth curve slopes and intercepts was modelled via random effects. Egg volume was significantly higher in the farmed population (60220.00 ± 7.62 $mm^3$) when compared with the wild (47623.78 ± 11.31 $mm^3$) population (Welsh two sample t test: df = 45.796, t = 9.4912, p < 0.0001). As such, direct inclusion of egg volume as a covariate to the growth models caused multicollinearity and problems with model convergence; hence, the potential effect of egg volume on morphometric parameters was tested for separately.

Variation in H/L ratio and the proportion of immature erythrocytes was analysed using a linear mixed effect model, where individual identifiers nested within cage identifiers were specified as a random effects to account for repeated measurement of the same individual. To achieve normality of model residuals, the H/L ratio was $\log_{10}$ scaled and the proportion of immature erythrocytes arcsine square root transformed. Effect of origin, sex, age and body mass, as well as all two-way interactions between these variables, were considered as predictors. Moreover, initial models on H/L ratio and the proportion of immature erythrocytes also

included three-way interactions between sex, age and body mass and between origin, age and body mass.

In the case of blood complement activity analysis, the half-life of bioluminescent bacteria exposed to mallard plasma (inversely related to complement activity) was used in linear mixed effect models as a response. Sex, origin, body mass and all two-way interactions between these variables were included as model predictors. Variation between cages was accounted for via random effects.

In all analyses, the minimal adequate model with all predictors significant ($p < 0.05$) was selected via step-wise elimination of nonsignificant terms from the initial full model. Random structure remained unchanged throughout the model selection process. An alternative statistical model, using random structure to control for the effect of sample location instead of cage effect, provided comparable results (not shown). The statistical package R 3.4.4 was used for all statistical calculations [57]. Raw data associated with all statistical analyses are provided in S1 Table.

## Results

We collected 37 eggs from wild population (four different localities) and 64 eggs from the two farms. A total of 69 eggs hatched successfully, comprising 26 wild (from three different localities) and 43 farmed ducklings. There was no significant difference in hatching success (Chi-squared test: df = 1, $\chi^2 = 0.0098$, $p = 0.9212$) between the wild and farmed populations. Nevertheless we only gathered data for 64 ducklings, two wild and three farmed ducklings being excluded to fit our experimental design (i.e. eight individuals per cage). The sex ratio did not differ between wild ($n_{males} = 13$, $n_{females} = 11$) and farmed ($n_{males} = 25$, $n_{females} = 15$) individuals included in the experiment (Chi-squared test: df = 1, $\chi 2 = 0.15547$ $p = 0.6934$).

### Growth rate and bill allometry

Variation of all morphometric parameters with age and in farmed vs. wild population is summarized in S1 Table. Both body mass and tarsus length showed a significant steeper gradual increase in the farmed population (Fig 2, S2A Table in S2 Table). And as a result, the average body mass and tarsus length differed significantly between farmed and wild 20-day-old ducklings (mean ± S.E: 321.6 ± 7.6 g, 46.4 ± 0.3 mm, and 238.9 ± 11.3 g and 42.2 ± 0.5 mm, respectively: t-test: $p < 0.0001$ in both cases). Conversely, structural body size (i.e. relative tarsus length) was higher in the wild population, and this difference was constant throughout the experiment. We also detected tarsus length increased more rapidly with age in males (as indicated by significant sex age2 interaction). But there were no sex-dependent differences in structural body size or body mass (Table 1). Indeed, egg volume was a strong predictor of body mass (slope = 1.4244 ± 0.3321 [estimate ± S.E.], $F_{(1,22)} = 18.39$, $p = 0.0003$), tarsus length (slope = 0.31051 ± 0.0980, $F_{(1,22)} = 10.05$, $p = 0.0044$) and structural body size (slope = -1.1139 ± 0.2506, $F_{(1,22)} = 19.76$, $p = 0.0002$) in 20-day-old ducklings in the wild population. At the same time, we detected a less pronounced effect of egg volume on body mass (slope = 0.1584 ± 0.1235, $F_{(1,38)} = 1.645$, $p = 0.2070$), tarsus length (slope = 0.0754 ± 0.0303, $F_{(1,38)} = 6.202$, $p = 0.0172$) and structural body size (slope = -0.0829 ± 0.1020, $F_{(1,38)} = 0.6611$, $p = 0.4212$) in the farmed population. Consequently, wild and farmed ducklings hatching from comparably sized eggs exhibited comparable tarsus length, body mass and structural size at 20 days (Fig 3). Models considering quadratic or asymptotic effects of egg volume did not explain the variation in wild population morphometric parameters any better than their linear counterparts ($p > 0.5$ in all cases).

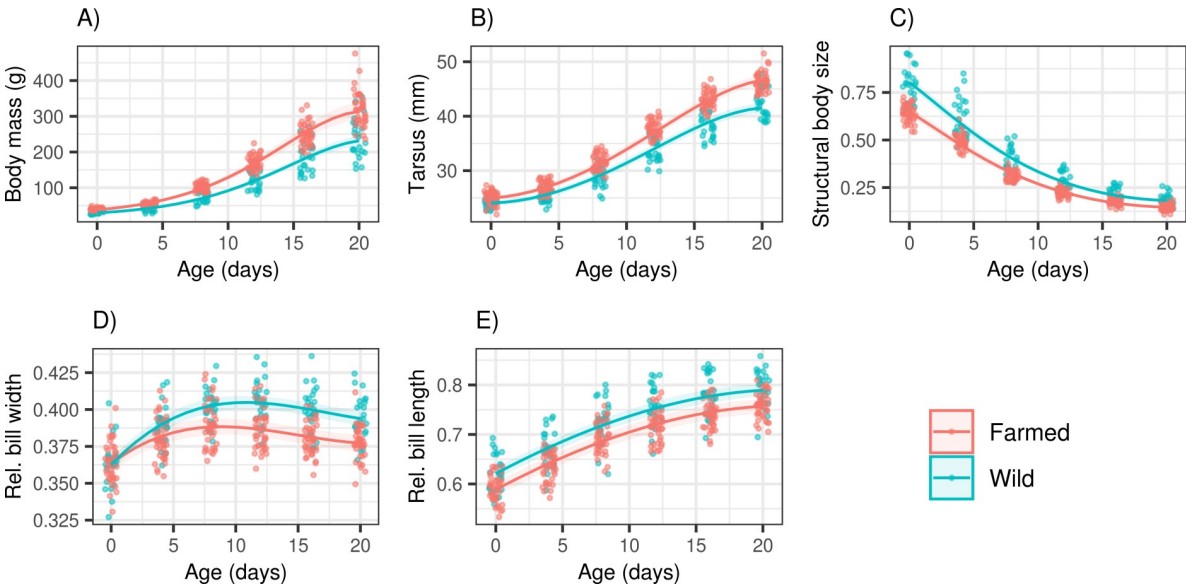

**Fig 2. Growth curves for five morphometric parameters.** (A) body mass, (B) relative body size, (C) structural body size (i.e. relative tarsus length), (D) relative bill width, (E) relative bill length in farmed and wild mallard populations ($n_{wild}$ = 24, $n_{farmed}$ = 40). Predictions are based on polynomial mixed effect models. The shaded area corresponds to 95% confidence intervals.

Growth models revealed that wild ducklings had longer and wider bills relative to actual tarsus length compared with their farmed counterparts and that these morphometric parameters did not differ between males and females. While the difference in bill length between farmed and wild individuals was constant throughout the common-garden experiment, the difference in bill width increased with increasing age (Fig 2, Table 1). Importantly, we observed no effect of egg volume on relative bill length (slope = -0.0273 ± 0.0509, F(1,38) = 0.2876, p = 0.5949) and width (-0.0016 ± 0.0256, F(1,22) = 0.0039, p = 0.9506) in the farmed population, and the same was true for wild individuals (bill length: -0.0889 ± 0.0694, F(1,22) = 1.64, p = 0.2137; bill width: 0.0952 ± 0.0687, F(1,38) = 1.9220, p = 0.1795; Fig 3).

## Haematology and complement activity

The H/L ratio was only affected by the body mass vs. age interaction (Fig 4, S2B Table in S2 Table, Table 2), the other predictors, including effect of origin, proved non-significant. Separate models for each age category showed a negative correlation between body mass and H/L ratio in 3-day-old ducklings (slope = -0.0035 ± 0.0009, $F_{(1,60)}$ = 16.1570, p = 0.0002), but no significant correlation in 9-day-old (slope = -0.0004 ± 0.0004, $F_{(1,60)}$ = 0.9935, p = 0.3229) and 15-day-old individuals (slope = -0.0003 ± 0.0004, $F_{(1,60)}$ = 0.7834, p = 0.3796).

A more complex variation pattern was observed in the case of immature erythrocytes, with proportions being affected by three-way interactions between body mass, age and origin (i.e. wild vs. farmed population; Fig 5) as well as a two-way interaction between sex and age (S2C Table in S2 Table, Table 3). Separate models for subsets of farmed and wild individuals indicated that farmed 3-day-old ducklings had higher proportion of immature erythrocytes than older age classes (p < 0.001 according to Tukey post-hoc tests), but no association between the proportion of immature erythrocytes and body mass (Δdf = 1, $\chi^2$ = 1.2648, p = 0.2607). On the other hand, there was an age-dependant relationship between the proportion of immature erythrocytes and body mass in wild individuals (age vs. body mass interaction: Δdf = 2, $\chi^2$ = 28.2250, p < 0.0001). Separate models for each age category indicated that immature

**Table 1. Parameter estimates for minimal adequate models describing variation of five morphometric parameters: Body mass, tarsus length, structural body size (i.e. relative tarsus length), relative bill length and relative bill width ($n_{wild} = 24$, $n_{farmed} = 40$).**

| Response | Predictor | Estimate | SE | DF | T | P |
|---|---|---|---|---|---|---|
| Body mass | Intercept | 2.0779 | 0.0094 | 220.1868 | 220.1868 | <0.001 |
| | Age | 0.8047 | 0.0134 | 59.8942 | 59.8942 | <0.001 |
| | Age$^2$ | -0.0673 | 0.0088 | 7.6343 | -7.6343 | <0.001 |
| | Age$^3$ | -0.0682 | 0.0057 | 12.0399 | -12.0399 | <0.001 |
| | Population (farmed vs. wild) | -0.1345 | 0.0149 | 9.0356 | -9.0356 | <0.001 |
| | Age: population (farmed vs. wild) | -0.0195 | 0.0177 | 1.1067 | -1.1067 | 0.2778 |
| | Age$^2$: population (farmed vs. wild) | 0.0266 | 0.0105 | 2.5355 | 2.5355 | 0.0171 |
| Tarsus length | Intercept | 1.5335 | 0.0041 | 371.4943 | 371.4943 | <0.001 |
| | Age | 0.2455 | 0.0041 | 60.0703 | 60.0703 | <0.001 |
| | Age$^2$ | -0.0112 | 0.0047 | 2.4054 | -2.4054 | 0.0318 |
| | Age$^3$ | -0.027 | 0.002 | 13.5276 | -13.5276 | <0.001 |
| | Population (farmed vs. wild) | -0.0396 | 0.0047 | 8.4069 | -8.4069 | <0.001 |
| | Sex (F vs. M) | 0.0024 | 0.0045 | 0.5398 | 0.5398 | 0.5912 |
| | Age: sex (F vs. M) | -0.0058 | 0.0043 | 1.3449 | -1.3449 | 0.1833 |
| | Age$^2$: sex (F vs. M) | 0.0051 | 0.0023 | 2.1981 | 2.1981 | 0.0314 |
| | Age: population (farmed vs. wild) | -0.0253 | 0.0049 | 5.1135 | -5.1135 | <0.001 |
| | Age$^2$: population (farmed vs. wild) | 0.0247 | 0.0051 | 0.0487 | 4.871 | <0.001 |
| Relative tarsus length | Intercept | -0.5421 | 0.0067 | 0.8100 | -80.9533 | <0.001 |
| | Age | -0.5654 | 0.0102 | 55.4054 | -55.4054 | <0.001 |
| | Age$^2$ | 0.0585 | 0.0049 | 11.9982 | 11.9982 | <0.001 |
| | Age$^3$ | 0.0412 | 0.0045 | 9.1067 | 9.1067 | <0.001 |
| | Population (farmed vs. wild) | 0.0927 | 0.0094 | 9.8749 | 9.8749 | <0.001 |
| Relative bill width | Intercept | -0.4206 | 0.0019 | 216.6831 | -216.6831 | <0.001 |
| | Age | 0.0093 | 0.0034 | 2.7548 | 2.7548 | 0.0173 |
| | Age$^2$ | -0.02 | 0.0021 | 9.5301 | -9.5301 | <0.001 |
| | Age$^3$ | 0.0069 | 0.0016 | 4.2884 | 4.2884 | 0.0019 |
| | Population (farmed vs. wild) | 0.0138 | 0.0031 | 4.3981 | 4.3981 | <0.001 |
| | Age: population (farmed vs. wild) | 0.0164 | 0.0048 | 0.0334 | 3.41 | 0.0022 |
| | Age$^2$: population (farmed vs. wild) | -0.009 | 0.0029 | 3.1053 | -3.1053 | 0.0081 |
| Relative bill length | Intercept | 0.6845 | 0.0065 | 105.992 | 105.992 | <0.001 |
| | Age | 0.137 | 0.0056 | 24.2662 | 24.2662 | <0.001 |
| | Age$^2$ | -0.0267 | 0.0035 | 7.5468 | -7.5468 | <0.001 |
| | Age$^3$ | -0.005 | 0.0027 | 1.8302 | -1.8302 | 0.0699 |
| | Population (farmed vs. wild) | 0.0324 | 0.0069 | 4.7108 | 4.7108 | <0.001 |
| | Sex (F vs. M) | 0.0105 | 0.0074 | 1.4301 | 1.4301 | 0.1576 |
| | Age: sex (F vs. M) | 0.0096 | 0.006 | 1.5997 | 1.5997 | 0.1149 |
| | Age$^2$: sex (F vs. M) | -0.009 | 0.0043 | 2.1207 | -2.1207 | 0.037 |
| | Age$^3$: sex (F vs. M) | 0.0073 | 0.0035 | 2.051 | 2.051 | 0.0425 |
| Relative bill width | Intercept | -0.4206 | 0.0019 | 216.6831 | -216.6831 | <0.001 |
| | Age | 0.0093 | 0.0034 | 2.7548 | 2.7548 | 0.0173 |
| | Age$^2$ | -0.0200 | 0.0021 | 9.5301 | -9.5301 | <0.001 |
| | Age$^3$ | 0.0069 | 0.0016 | 4.2884 | 4.2884 | 0.0019 |
| | Population (farmed vs. wild) | 0.0138 | 0.0031 | 4.3981 | 4.3981 | <0.001 |
| | Age: population (farmed vs. wild) | 0.0164 | 0.0048 | 0.0334 | 3.41 | 0.0022 |
| | Age$^2$: population (farmed vs. wild) | -0.009 | 0.0029 | 3.1053 | -3.1053 | 0.0081 |

Age predictor was modelled as linear (Age), quadratic (Age2) and cubic (Age3).

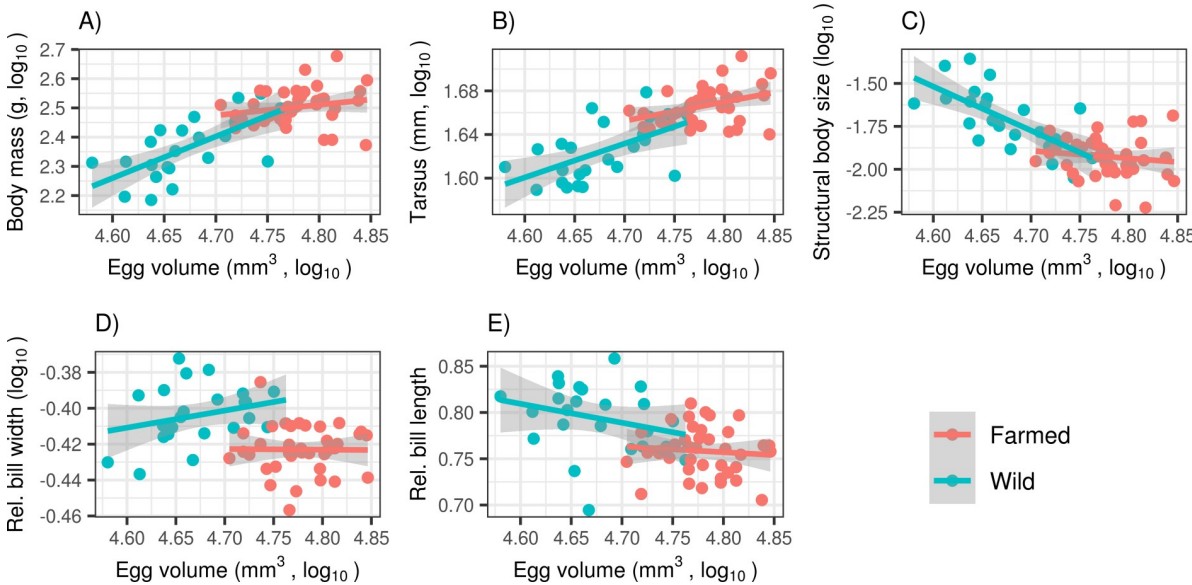

**Fig 3. Correlation between egg volume and five morphometric parameters for 20-day-old (i.e. terminal stage of the experiment).** (A) body mass, (B) relative body size, (C) structural body size (i.e. relative tarsus), (D) relative bill width, (E) relative bill length in farmed and wild mallard populations ($n_{wild}$ = 24, $n_{farmed}$ = 40). Predictions are based on linear regression and shaded areas correspond to 95% confidence intervals.

erythrocytes were positively correlated with body mass in 3-day-old wild ducklings (slope = 0.0018 ± 0.0004, $F_{(1,21)}$ = 19.9160, p = 0.0002), but not in 9-day-old (slope = -0.0001 ± 0.0002, $F_{(1,21)}$ = 0.5083, p = 0.4837) or 15-day-old ducklings (slope = -0.0002 ± 0.0001, $F_{(1,21)}$ = 3.5060, p = 0.0751). Notably, there was a negative correlation between proportions of immature erythrocytes in 3-day-old vs. 9-day-old wild individuals (Spearman's correlation: rho = -0.4191, p = 0.0426) and similar non-significant association was observed also between 3-day-old and 15-day-old wild ducklings (Spearman's correlation: rho = -0.3546, p = 0.0890). At the

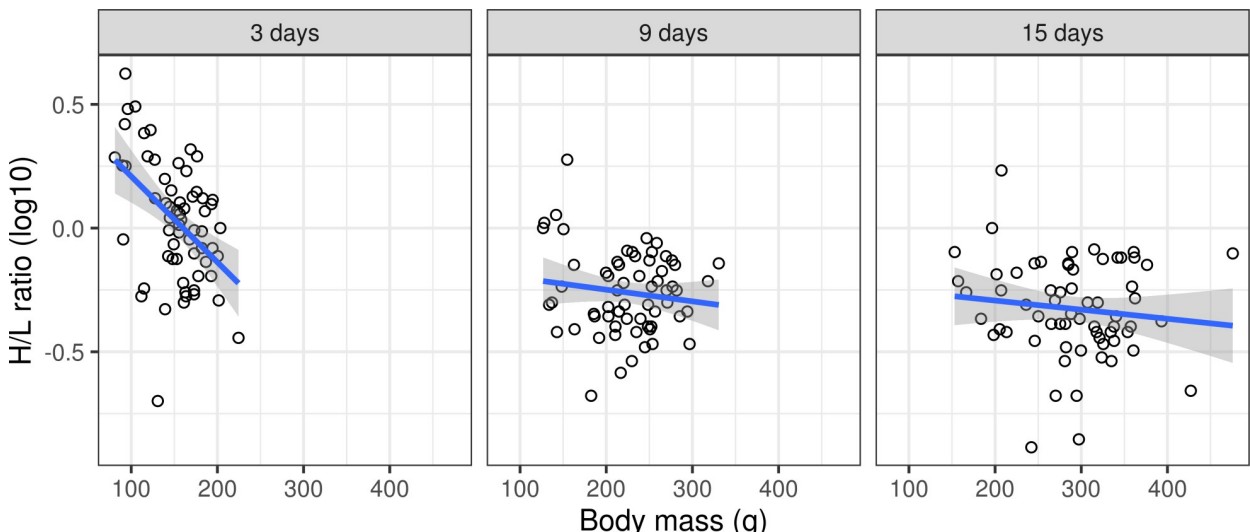

**Fig 4. Variation in mallard heterophil vs. lymphocyte (H/L) ratio due to the effect of body mass (n = 64).** Predictions are based on mixed models and shaded areas correspond to 95% confidence intervals.

**Table 2. Parameter estimates for minimal adequate models on the heterophil vs. lymphocyte (H/L) ratio ($n_{wild}$ = 24, $n_{farmed}$ = 40).**

| Predictor | Estimate | SE | DF | T | P |
|---|---|---|---|---|---|
| Intercept | 0.575 | 0.121 | 101.831 | 4.754 | <0.001 |
| Body mass | -0.004 | 0.001 | 105.791 | -4.628 | <0.001 |
| Age (4 days vs. 9 days) | -0.738 | 0.135 | 123.490 | -5.465 | <0.001 |
| Age (4 days vs. 15 days) | -0.809 | 0.133 | 124.572 | -6.085 | <0.001 |
| Body mass: age (4 days vs. 9 days) | 0.003 | 0.001 | 135.683 | 4.176 | <0.001 |
| Body mass: age (4 days vs. 15 days) | 0.003 | 0.001 | 152.011 | 4.499 | <0.001 |

3-day-old individuals, the linear association of response with body mass provided a more parsimonious fit than either quadratic (F(1,21) = 0.1015, p = 0.7532) or asymptotic (ΔAIC = 1.85) associations. Moreover, after controlling for the effect of body mass and origin, there was an increase in the proportion of immature erythrocytes in 3-day-old males compared with females ($F_{(1,58)}$ = 4.7687, p = 0.0330), but no sex-dependent differences in 9-day-old ($F_{(1,58)}$ = 1.6503, p = 0.204) or 15-day-old ($F_{(1,58)}$ = 0.7464, p = 0.3911) individuals.

When omitting the effect of other variables, bioluminescent bacterial half-life was higher when exposed to wild mallard plasma than farmed mallard plasma, indicating compromised complement activity in the former group (Δdf = 1, $\chi^2$ = 8.523, p = 0.0035, S2D Table in S2 Table). More complex models, including the effect of sex, body mass and their interactions, revealed that plasma complement activity showed contrasting co-variation with body mass in farmed and wild individuals (Fig 6, Table 4). Specifically, we observed a non-significant positive correlation with body mass in wild individuals (slope = 0.6952 ± 0.3903, $F_{(1,21)}$ = 3.1730, p = 0.0893), indicating decreased complement activity in individuals of higher body mass, and a non-significant negative association in farmed individuals (slope = -0.5766 ± 0.3301, $F_{(1,36)}$ = 3.0510, p = 0.0892).

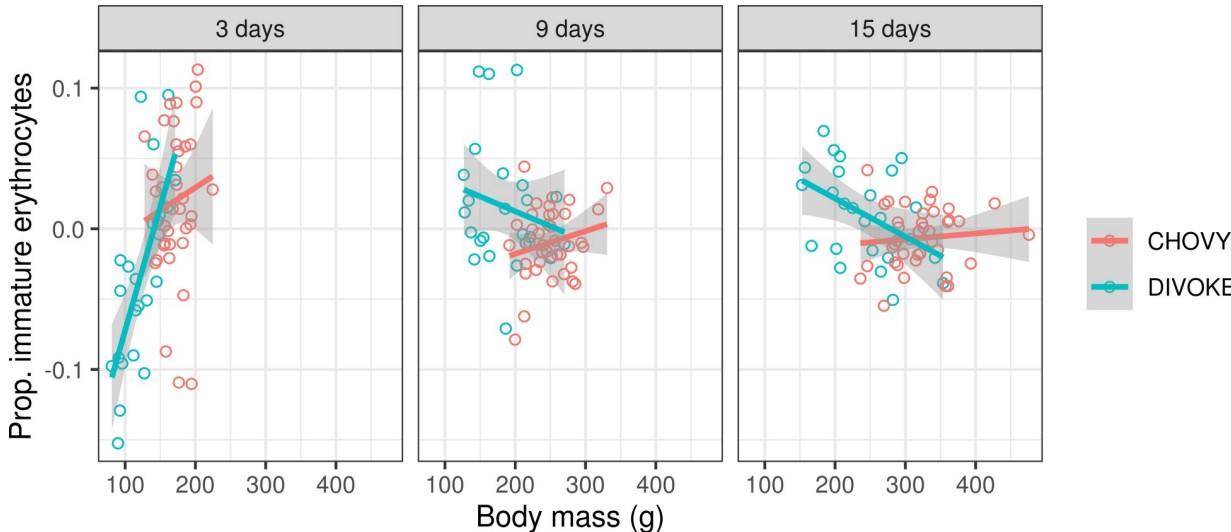

**Fig 5. Variation in the proportion of immature erythrocytes in wild and farmed mallard, including variation due to the effects of body mass and age ($n_{wild}$ = 24, $n_{farmed}$ = 40).** Analysis was adjusted for other variables present in the minimal adequate model. Individual observations correspond to model residuals. Model predictions and 95% confidence intervals are shown.

**Table 3. Parameter estimates for the minimal adequate model for proportion of immature erythrocytes ($n_{wild} = 24$, $n_{farmed} = 40$).**

| Predictor | Estimate | SE | DF | T | P |
|---|---|---|---|---|---|
| Intercept | 0.135 | 0.053 | 173.693 | 2.547 | 0.012 |
| Body mass | 0.000 | 0.000 | 170.680 | 1.048 | 0.296 |
| Sex (F vs. M) | 0.029 | 0.010 | 172.041 | 2.829 | 0.005 |
| Population (farmed vs. wild) | -0.220 | 0.065 | 174.352 | -3.366 | 0.001 |
| Age (3 days vs. 9 days) | 0.068 | 0.073 | 169.913 | 0.930 | 0.354 |
| Age (3 days vs.15 days) | 0.025 | 0.068 | 170.394 | 0.369 | 0.713 |
| Body mass: age (3 days vs. 9 days) | 0.000 | 0.000 | 170.142 | -0.409 | 0.683 |
| Body mass: age (3 days vs. 15 days) | 0.000 | 0.000 | 170.484 | -0.822 | 0.412 |
| Sex (F vs. M): age (3 days vs. 9days) | -0.042 | 0.014 | 168.435 | -2.931 | 0.004 |
| Sex (F vs. M): age (3 days vs. 15 days) | -0.036 | 0.014 | 168.448 | -2.556 | 0.011 |
| Population (farmed vs. wild): age (3 days vs. 9 days) | 0.327 | 0.089 | 169.536 | 3.674 | <0.001 |
| Population (farmed vs. wild): age (3 days vs. 15days) | 0.322 | 0.085 | 170.285 | 3.789 | <0.001 |
| Body mass: population (farmed vs. wild) | 0.001 | 0.000 | 171.497 | 3.433 | 0.001 |
| Body mass: population (farmed vs. wild): age (3 days vs. 9 days) | -0.002 | 0.001 | 169.699 | -3.670 | <0.001 |
| Body mass: population (farmed vs. wild): age (3 days vs. 15days) | -0.002 | 0.000 | 170.548 | -3.862 | <0.001 |

## Discussion

This is the first study to compare morphometric and allometric data between wild and farmed mallard populations using a common-garden experiment. Most previous works studying the morphological parameters of wild vs. farmed mallards have only included full-grown individuals exposed to different conditions their whole lives (e.g. [14,25,27,29]). Furthermore, this is the first study to also compare variation in haematological parameters and immune function between wild vs. farmed mallard populations using this model system.

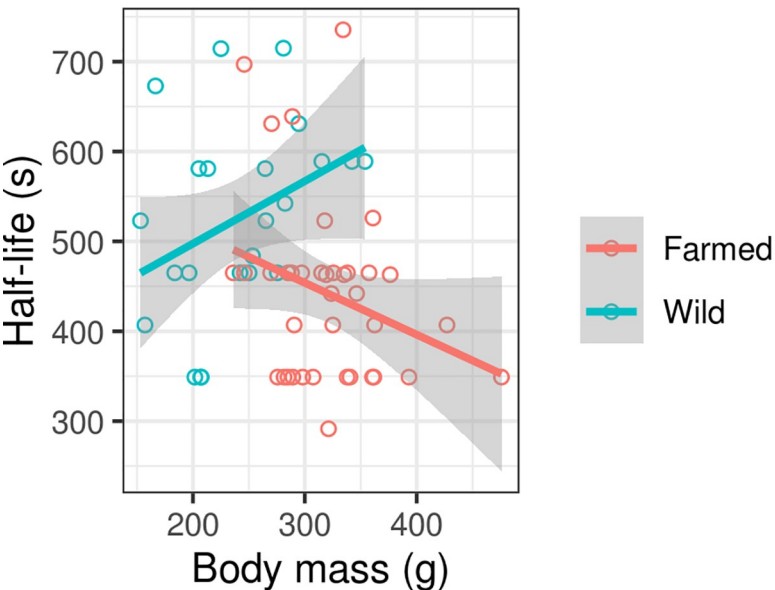

**Fig 6. Variation in bioluminescent bacterial half-life (inversely related to complement activity) following exposure to 20-day-old mallard plasma (n = 61).** Predictions are based on mixed models and shaded areas correspond to 95% confidence intervals.

**Table 4. Parameter estimates from the minimal adequate model for bioluminescent bacterial half-life following exposure to mallard plasma, used as a proxy of complement activity ($n_{wild}$ = 23, $n_{farmed}$ = 38).**

| Predictor | Estimate | SE | DF | T | P |
|---|---|---|---|---|---|
| Intercept | 626.889 | 108.951 | 56.943 | 5.754 | <0.001 |
| Body mass | -0.577 | 0.335 | 56.943 | -1.719 | 0.091 |
| Wild population | -268.374 | 143.829 | 56.943 | -1.866 | 0.067 |
| Body mass: wild population | 1.272 | 0.507 | 56.943 | 2.508 | 0.015 |

We are aware that our experimental setup does not directly control for potential variation introduced by maternal effects, a confounding factor that could theoretically be suppressed by using F1 generation individuals raised under experimental conditions. In this case, however, selection may already have acted on the parental generation, which would cause unpredictable bias in the experimental output. Further, as farmed and wild populations may adapt differently to the conditions in our experimental environment, maternal effect could still play a role, even on the F1 generation [58].

In our experiment, farmed ducklings exhibited higher growth rates for both body mass and tarsus. Surprisingly, our findings are similar to those from a 50-year-old study from Northern America [33], where farmed ducklings were only reared in captivity for a few generations. The higher growth rates observed in farmed individuals may be partly explained by different environmental conditions under captivity (i.e. high food availability and predator absence), which generally favour individuals allocating more energy to growth and reproduction than to other physiological and immune functions [59]. In addition, there is evidence for past hybridisation between farmed mallards and domestic duck strains with intentional artificial selection for phenotypes [18,22]. In addition to differences in growth rate, Prince *et al.* [33] also observed sex-specific differences; however, the effect of sex in our own study was rather low. We believe that this discrepancy could have been caused by the shorter duration of our experiment.

In contrast to body mass and tarsus length, structural body size was consistently higher in wild ducklings throughout the experimental period. However, this finding also suggests reduced mass of non-skeletal tissues in wild population, which is commonly interpreted as decreased body condition in ecological studies. Thus our data suggest that farmed individuals may allocate more alimentary resources to non-skeletal tissues under ad libitum feeding regime compared with those from wild populations. Alternatively, there might be stronger selection for larger structural body size under natural conditions as movement ability is essential for precocial juveniles during the post-hatching period [60]. It can also mean that body mass is more affected than tarsus length by a limitation of food under natural conditions.

Maternal investment, measured as egg volume, was significantly lower in the wild population in our study, while wild morphometric parameters showed greater co-variation with increasing maternal investment at 20-days-old. As a consequence, we cannot exclude that the observed differences in growth rates were driven, to some degree, by lower maternal investment in the wild population. Nevertheless, the contrasting effects of maternal investment on growth rates for the two populations, as well as the inability of wild ducklings to compensate for these relatively small differences after three weeks under *ad libitum* conditions, deserves further attention. One can argue that maternal effect may only be manifested if investments into the egg are suboptimal and that this only applies in wild populations. If this were true, the strength of correlation between morphometric parameters and egg volume should increase with decreasing egg volume; or, in other words, there should be a nonlinear asymptotic association instead of a linear association between egg volume and morphometric parameters in wild populations. However, our data does not support this possibility. As an alternative

explanation, we propose that there may be higher genetically-determined covariance between growth rates during early ontogeny in wild populations and later energetic investment into the clutch [61]. This effect may play a lesser role in farmed populations due to the genetic homogeneity of breeding stocks [22]. Maternal investment in wild populations may also reflect actual foraging opportunities at the breeding site and, at the same time, adaptive modulation of hatchling growth rate [62]. On the contrary, maternally-induced delayed growth is unlikely to be adaptive in farmed populations exposed to an *ad libitum* food supply for many generations.

In our study, wild ducklings exhibited longer and wider bills relative to their body size than farmed ducklings of comparable body size, and further analysis indicated that this difference was unlikely to arise as a consequence of maternal investment variation. Hence, in line with previous studies on mallards [27,29] and other bird taxa (e.g. [63]), our data suggest that bill morphology is relatively plastic and can rapidly respond to different selection pressures in wild and captive environments. Specifically, altered food composition in breeding facilities (e.g. wheat and maize grains, food-pellets) could lead to selection for a bill shape that is less adapted for effective harvesting of water invertebrates, which constitute the essential protein source during early post-hatching stages in wild populations [27]. Moreover, while not analysed in our study, a lower lamellar density has also been reported on the bill filtering apparatus of wild populations [27,29]. Unlike the latter two studies, we detected no sex-specific differences in bill morphology, though this may have been partly caused by our focus on juvenile individuals. In contrast to our data, Söderquist *et al.* [29], studying historical changes in bill morphology, found a decrease in bill width in wild populations following the establishment of massive restocking, probably due to introgression of alleles affecting bill shape. It should be mentioned this study focused only on historical phenotype shifts. As mallards were not reared in the same environment, other factors, such as habitat and climatic changes, could contribute to these changes.

A higher H/L ratio is frequently used as an indicator of stress [34] and/or disease [64]. We observed a strong negative correlation between body mass and H/L ratio in 3-day-old ducklings, with the correlation disappearing with increasing age. Consequently, we argue that such rapid changes in H/L ratio variation put the general usefulness of this index as a stress indicator into question, at least for the juvenile cohort. Furthermore, our data did not support the prediction that H/L ratio would be lower in farmed populations as a consequence of a) adaptation to captive conditions (resembling conditions in breeding facilities), and b) a general decrease in physiological stress responses in farmed individuals [65–67]. In fact, when accounting for differences in body mass between the two groups, H/L ratio was the same for farmed and wild individuals.

In this study, we also analysed the proportion of immature erythrocytes, an indicator of haematopoiesis rates linked with resistance to anaemic diseases [35] frequently induced by environmental stress in free living populations [68–70]. We found that the proportion of immature erythrocytes varied with body mass in a contrasting manner in farmed and wild populations. Three-day-old farmed ducklings had an increased proportion of immature erythrocytes compared to older individuals, indicating accelerated haematopoiesis rates shortly after hatching. Furthermore, we found that the proportion of immature erythrocytes varied with body mass and age in a contrasting manner in farmed and wild populations. There was no association with body mass in farmed population. On the other hand, a positive correlation with body mass was observed in the case of 3-day-old wild ducklings, suggesting that wild individuals with superior body condition could afford accelerated haematopoiesis. Nevertheless, compared to 3-day-old wild ducklings with high immature erythrocytes levels, wild ducklings with low immature erythrocytes levels at this age exhibited comparative increase of haematopoiesis (i.e. higher proportion of immature erythrocytes proportions) during later

developmental stages. Consequently, positive association with body mass disappeared in 9-day-old and 15-day-old wild individuals. Instead, we observed a non-significant negative correlation with body mass in these age classes. It is tempting to speculate that the contrast shown between wild and farmed populations, specifically during the third day of life, may be associated with lower maternal investment into the eggs in wild populations. However, as with the analysis of growth parameters, the haematopoiesis vs. body mass association tended to be linear rather than quadratic or asymptotic, as would be expected if maternal effects only played a role in a subset of individuals where energetic investment into the egg was suboptimal. Hence, we believe that the contrast between wild and farmed individuals is unlikely to be explained by differences in maternal investment only.

In this study, not only was total complement activity lower in the wild population but plasma complement activity tended to increase with body mass in farmed ducklings and decrease in wild ducklings. This pattern did not fit with our expectation that plasma complement activity would be lower in farmed ducklings due to a lower pathogen burden. Differences in complement activity between the two populations are also unlikely to be explained by the effects of stress as all individuals were subjected to the same amount of human disturbance and there was no difference in H/L ratio between the wild and farmed populations. Instead, we suggest that limited and unpredictable food availability in wild populations may select for conservative investment of energetic resources to costly physiological and immune functions, while farmed populations are largely released from such trade-offs [71,72]. There have been relatively few studies comparing immune function in wild and farmed bird populations. Nevertheless, our results are not fully consistent with Buehler *et al.* [40] and Homberger *et al.* [73], who found no difference in innate immunity between farmed and wild populations of red knot (*Calidris canutus*) and grey partridge (*Perdix perdix*), respectively (though it should be noted that our study used a different methodology for measuring innate immunity).

In conclusion, using a common-garden experiment, we recorded differences at morphological, haematometric and immunological levels between wild mallard ducklings and farmed individuals released in great numbers for hunting purposes. From conservation point of view, massive introduction of phenotypes that are distinct from those present in native populations is always highly controversial practice. Our study therefore provides another piece of evidence that current restocking of mallards is insufficiently managed and may induce undesired effects on native populations. Importantly, wild ducklings had longer and wider bill relative to their body size. As bill morphology is to large extent determined by foraging niche, further research should explore consequences of this morphological divergence on food collection efficiency under natural conditions. Farmed individuals also exhibited more rapid growth and haematopoiesis shortly after the hatching, as well as higher complement activity. These differences may indicate superior performance of farmed population under conditions of our experiment. However, as already shown by previous studies on various animal species (e.g. [12,16]), strains selected for good performance under captivity typically exhibited decreased fitness, if were not exposed to various stressors present in natural environment. This is also consistent with low survival rates [14,23,24] and low level of genetic introgression [21,22] of farmed mallards in native populations. Altogether, phenotype differences documented by both our study and previous studies highlight the potential risk for phenotypic shift and a subsequent effect on the fitness of wild populations exposed to massive restocking. More proper monitoring of restocking activities and management of facilities producing farmed individuals (e.g. food composition similar to wild population, genetic assessment of released birds) would be desirable in order to limit phenotypic and genotypic diversification between wild and captive populations [74]. Our results can be useful also in wider perspective because other game bird species such

as pheasant, grey partridge, red-legged partridge or mammalian and fish species are massively released throughout the world.

## Supporting information

**S1 Checklist.**
(PDF)

**S1 Table. Descriptive statistics for morphometric parameters and raw data on morphometric and haematological analyses and on analyses of complement activity.**
(XLSX)

**S2 Table. Tables describing model selection for each response variable.** Support for predictors was tested using likelihood-ratio tests assuming $\chi^2$ distribution of deviance changes. Order of elimination of nonsignificant predictors is shown in corresponding column. Predictors included in the minimal adequate model (either as a main effect or through interaction with other variables) are in bold face. S2A Table. Model selection process for body mass, tarsus length, relative tarsus length, relative bill width and length. S2B Table. Model selection process for H/L ratio. S2C Table. Model selection process for proportion of immature erythrocytes. S2D Table. Model selection process for complement activity.
(XLSX)

## Acknowledgments

We thank Dagmar Čížková for her help with field work, Jocelyn Champagnon and Pär Söderquist for their helpful comments. We are also grateful to Oto Čížek for his help with the administrative procedures required for egg collection in wild populations and to Michal Vinkler for explaining erythrocyte analysis.

## Author Contributions

**Conceptualization:** Jana Svobodová, Hana Pinkasová, Jakub Kreisinger.

**Data curation:** Jakub Kreisinger.

**Formal analysis:** Hana Pinkasová, Jakub Kreisinger.

**Funding acquisition:** Jakub Kreisinger.

**Investigation:** Jana Svobodová, Hana Pinkasová, Pavel Hyršl, Monika Dvořáčková, Lukáš Zita, Jakub Kreisinger.

**Methodology:** Jana Svobodová, Hana Pinkasová, Jakub Kreisinger.

**Project administration:** Jana Svobodová, Hana Pinkasová, Jakub Kreisinger.

**Resources:** Jakub Kreisinger.

**Software:** Jakub Kreisinger.

**Supervision:** Jakub Kreisinger.

**Validation:** Jakub Kreisinger.

**Visualization:** Jakub Kreisinger.

**Writing – original draft:** Jana Svobodová, Jakub Kreisinger.

**Writing – review & editing:** Jana Svobodová, Hana Pinkasová, Pavel Hyršl, Jakub Kreisinger.

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
