## [Decision Letter · Decision Letter 0]

1 May 2020

PONE-D-20-03121

Differences in the growth rate and immune strategies of farmed and wild mallard populations

PLOS ONE

Dear Dr. Svobodová,

Thank you for submitting your manuscript to PLOS ONE. After careful consideration, we feel that it has merit but does not fully meet PLOS ONE’s publication criteria as it currently stands. Therefore, we invite you to submit a revised version of the manuscript that addresses the points raised during the review process.

Both reviewers have performed an exhaustive revision of the manuscript, and it would be very good for the manuscript improvement that you follow their suggestions.

We would appreciate receiving your revised manuscript by 1 month. To enhance the reproducibility of your results, we recommend that if applicable you deposit your laboratory protocols in protocols.io, where a protocol can be assigned its own identifier (DOI) such that it can be cited independently in the future. For instructions see: http://journals.plos.org/plosone/s/submission-guidelines#loc-laboratory-protocols

We look forward to receiving your revised manuscript.

Kind regards,

Magdalena Ruiz-Rodriguez

Academic Editor

PLOS ONE

2. In your Methods section, please provide additional information regarding the permits you obtained for the work. Please ensure you have included the full name of the authority that approved the collection sites access and, if no permits were required, a brief statement explaining why.

3. As part of your revision, please complete and submit a copy of the ARRIVE Guidelines checklist, a document that aims to improve experimental reporting and reproducibility of animal studies for purposes of post-publication data analysis and reproducibility: https://www.nc3rs.org.uk/arrive-guidelines. Please include your completed checklist as a Supporting Information file. Note that if your paper is accepted for publication, this checklist will be published as part of your article."

4. In your Methods section, please include a comment about the state of the animals following this research. Were they euthanized or housed for use in further research? If any animals were sacrificed by the authors, please include the method of euthanasia and describe any efforts that were undertaken to reduce animal suffering."

Reviewers' comments:

Reviewer's Responses to Questions

**Comments to the Author**

1. Is the manuscript technically sound, and do the data support the conclusions?

Reviewer #1: Partly

Reviewer #2: Yes

2. Has the statistical analysis been performed appropriately and rigorously? 

Reviewer #1: Yes

Reviewer #2: I Don't Know

3. Have the authors made all data underlying the findings in their manuscript fully available?

Reviewer #1: No

Reviewer #2: No

4. Is the manuscript presented in an intelligible fashion and written in standard English?

Reviewer #1: Yes

Reviewer #2: Yes

5. Review Comments to the Author

Reviewer #1: SUMMARY

This article addresses the question of the low survival in the wild of captive-bred mallards released for hunting. It compares the early duckling’s development (over the first 20 days) of wild and captive-bred mallards in a control captive experiment, focusing on morphological, stress and immune parameters.

While this topic is interesting and the study design appropriate, the article presents some mistakes and the conclusion is not totally supported by the results. I suggest below some revisions to meet the journal's requirements.

MAJOR ISSUES

First, some claims are not sufficiently justified, in particular the conclusions (lines 432-439)

The results suggest an appropriate response of captive-bred mallards to survive in the wild. With higher maternal investment, captive-bred ducklings had a growth rate, haematopoiesis and immune system that looks rather suitable to survive in a wild environment. Nevertheless, the authors concluded to a divergence with wild mallard that may contribute to the lower survival of captive-bred mallards in a natural environment. Because the results did not totally support this statement, the hypotheses and mechanisms explaining the claim should be detailed.

In addition, even if the results suggest that the populations are genetically divergent, this result has been already shown in other studies (through more robust analysis of population genetics) and by focusing on early development, this study does not contribute with new information to claim that massive restocking could lead to a decrease in fitness of wild populations (last sentence in the abstract and conclusions). Here also, the authors must argue their point through the description of the mechanism, based on their results.

Second, I identified some errors that could be typos but could also reflect more serious concerns on the text-to-figures matching.

First the sample size in the methods: In the text, line 117-123, it is written that 26 and 43 eggs were collected in the wild and in hunting farms respectively, but it contradicts the figure 1 (37 and 64 eggs were collected respectively). From the results, I understood that the numbers presented in the methods refer to the numbers of eggs hatched and not collected.

Second and more importantly, the results presented in line 264-265 contradicts the figure 3. The slopes in the text are negatives while in the figures 3D and 3E, the slopes of wild mallard subset look clearly positives.

Finally, all the tables, figures and supplementary materials present an error on the word "tarsus" written "tarzus"

Third, the method section is not sufficiently detailed on the ethics because it did not describe the fate of the ducklings after 20 days. Were they sacrificed? Journal requires that if anaesthesia, euthanasia, or any kind of animal sacrifice is part of the study, the article should include briefly which substances and/or methods were applied.

Fourth, the authors stated that “all relevant data are within the manuscript and its supporting information files”. Nevertheless, only the results of the model are available in the text and the supplementary materials, not the raw data.

MINOR ISSUES

line 63-65: It is not appropriate to make a numerical comparison between the number of European breeding pairs and the number of juveniles released in the region, as these breeding pairs can potentially give birth to more than 10-12 ducklings each. In addition, the pairs breed in spring while the juveniles are released in summer, primarily at the onset of the hunting season.

line 125: Please specify whether the collection of eggs in captivity took place from newly laid eggs. If not please detail the age of the eggs as presented for the collection of wild eggs.

line 354-357: This argument needs to be more cautious. The lack of detection of a non-linear asymptotic relationship may be due to a lack of statistical power, the linear model being more parsimonious in terms of number of parameters.

OTHER COMMENTS:

line 141: Were the ducklings marked at the age of one day? Please specify.

line 238: "a significant increase" is not clear because both populations showed it. Authors probably mean that "Body mass[...] showed a significant higher increase in the farmed population [...]".

line 324: The same problem as above arises and I suggest replacing "increased" with "higher" here.

Supplementary materials: I found the presentation of the tables without any text rather sketchy. I suggest also to highlight the significant results in the tables. In addition, the predictors probably refer to different models but only the P values are presented. I suggest to include model selection with deviance, etc.

Reviewer #2: Restocking wild populations with captive-bred individuals is common not only to help threatened populations but also to boost huntable populations and increase hunting bags, which is the case in Europe for the mallard. Earlier studies have shown both morphological and genetical differences between these released individuals and their free-living conspecifics. The two groups are also hybridizing resulting in potential negative effects for the wild population. The present study aims to growth-parameters as well as immune functions in both captive-bred and wild individuals in a common-garden experiment. Their results show differences in both morphology, growth-rates and immune functions and are explained by both genetical and environmental factors.

It is an interesting and important study that contribute to the understanding of the effects of large-scale releases of captive-bred individuals. Although it deals with questions that, at least partly, already have been studied in other articles, I believe that it adds to research area with a smart and simple experimental set-up that focus on the important period of a captive-bred ducklings’ life when it is in the hands of humans.

The manuscript is well-written and relatively easy to take in. The authors should however try to elaborate on their findings, specially about the immune functions and also put them in a wider perspective.

Comments below reference to the line numbers in the manuscript.

Abstract

27: Change “anatomical” to “morphological”.

29: Change “in free-living populations and those from breeding facilities” to “between free-living populations and individuals from breeding facilities”.

34: Maybe “greater” is more correct than “higher”?

38: If the result is negative maybe “due to” is more appropriate.

Introduction

61: Delete “on” and “of” (before “duck”).

62: Change “stocking” to “restocking”.

63: Maybe worth mentioning somewhere, the extent of releases in Czech Republic?

63: In the manuscript you use several different terms for these mallards: Captive-bred, captive-reared, farmed etc. And there exists many more in the different articles you cite. Maybe worth mentioning that several different terms are used (sometimes with different meanings) and that you in this manuscript use the following… Stick to one or two and be consistent. It could also be good to state what you mean with the terms you use.

70: Add reference.

79: Full stop after references. Change next sentence to: “However, the phenotypic changes observed in such cases will not necessarily have been caused by restocking alone, but also other factors, such as habitat and climate change, may be of comparable importance [30].”

84: Delete “there”.

88: Change to: “(up to an age of 20 days)”.

90: Delete extra space after references.

102-105: I think you need to be clearer here about what you do and what you are testing. I am not sure what prediction you are testing here. When you rear them both in captivity and predict that wild eggs hatched and reared in captivity will have a lower growth and haematopoiesis compared to captive-reared, you at the same time predict that this is genetically determined, right?

Methods

116-117: You collected 37 wild eggs from four localities. If you only want to include the hatched eggs, you should also mention that they are only from three localities. But I think that is better to mention in the results.

121: You mean less than (<) four days?

122-123: You collected 64 farmed eggs.

141: Delete extra space after references.

142: At the end of the sentence, add: “in each cage”.

150: Rephrase sentence using passive form and avoid pronouns.

152-153: With an accuracy of 0.01mm?

152: How was the bills measured? Over the nostrils? Describe and possibly give a reference.

153-154: Why not sample them the same days as you measured them?

153-155: Type of syringe and size of needle?

177: Rephrase sentence and avoid using pronouns.

188-189: Have you considered using the residuals of a body mass and tarsus length/bill measurements regression? This is common when using body condition indices. See e.g:

Green, A.J. (2001). Mass/length residuals: measures of body condition or generators of spurious results? Ecology, 82, 1473-1483.

Jakob, E.M., Marshall, S.D. & Uetz, G.W. (1996). Estimating Fitness: A Comparison of Body Condition Indices. Oikos, 77, 61-67.

Schulte-Hostedde, A.I., Zinner, B., Millar, J.S. & Hickling, G.J. (2005). Restitution of mass–size residuals: validating body condition indices. Ecology, 86, 155-163.

In any way, wouldn’t body mass divided by size be a more appropriate way to calculate the variable?

201: You could only measure to nearest 0.01mm, so four decimals feels a bit too much.

209: Change to: “specified as random effects…”

223-228: I assume you use a 0.05 significance level, maybe best to state that clearly.

229: I am not sure on what level raw data should be uploaded/available for others but a supplementary file with all measurements and test values could be of interest for other researchers.

Results

231: How many males and females in each group? Was the sex ratio equal in both groups?

238-241: You mean a significant higher body mass and tarsus length in farmed compared to wild? When you write increase it sounds like you are talking about growth rate. I guess you tested the differences with a t-test or similar? Please give p-values and test-values.

241-243: You give the measurements after 20 days and conclude that relative tarsus length was higher in the wild after 20 days. And then state that it remained higher throughout the experiment. But the experiment was over after 20 days? So, maybe rephrase by writing that it was higher directly after hatching and remained so after 20 days. Or simply change “remained” to “was”

243-244: Mild effect? If I look at the correct values, the p-value is 0.59. I would not call that a mild effect. Or should it be 0.059? Or do you mean in age2 but not in the other age-groups? Age2 is significant so no need to call it a mild effect.

244: Change “then” to “than”. It is a bit confusing when you sometimes use structural body size and sometimes relative tarsus length.

245-255: Sounds like discussion.

259-260: The parenthesis is quite important because I don’t know if the actual bill length or width differ between wild and farmed in your study. There is no data that shows the actual measurements of the bills. I think that should be included, at least in the supplemental information.

261-263: Again, did it differ? Or do you mean then relative sizes?

268-269: Discussion.

274: Change “association” to “correlation”.

282: Change ”an increased” to “a higher”.

294: Delete “slight”. There was a significant difference according to your test.

Discussion

337-338: I would argue that wild mallards have a lower weight in relation to their tars, i.e. they have a lower body condition, therefore, I would divide weight with the size variable (or preferably calculate body condition by using the residuals of a body mass and tarsus length regression, as proposed above).

338-343: Well, the tarsus is still shorter in wild mallards. What you write in 342-343 is probably correct and would therefor also mean that body mass is probably more affected than tarsus length under natural conditions with a limitation of food.

340: “Structural tarsus length” is not a term you have used before. You mean relative tarsus length i.e. structural body size?

367: Is this correct? Or do you again mean relative sizes? The actual sizes don’t have to be longer just because the relative sizes are. Again, it would be interesting to see the numbers on the actual sizes. You also state that wild had wider bills than farmed but further down (line 376) you are talking about a decrease. Because you have not compared mallards from different time periods, I think you should be cautious to use phrases like increased or decreased when you just mean that your two groups are different to each other. You can talk about increased if you have a baseline or compare the same variable over time. But here you often use it when one group have a higher value than the other.

375-377: No, that is not correct. According to Söderquist et al. 2014, historical wild mallards had the narrowest bills while farmed had the widest, placing contemporary wild mallards in the middle. So, wild mallards showed an increase in bill width since the start of releases of farmed mallards.

379: Change “lamella” to “lamellar”.

385: Change “association” to “correlation”.

390: delete extra space before parenthesis.

400: Chane “than” to “compared to”.

403-406: I find this a bit confusing. There was a positive correlation in 3-day-old, the higher weight, the higher production of erythrocytes. But the correlation changed to negative in 9- and 15-day-old. In 9- and 15-day-old the light individuals have the highest production of erythrocytes. And it is these individuals that you mean compensate for low production during earlier stages? Because the light ones in 9- and 15-day-old were probably also light in 3-day-old, i.e. they had a low production at that stage? Consider rewriting this part to make it more clear which parts of the graphs you are talking about.

415: Change “decreased” to “lower”.

418: Change “lowered” to “lower”.

435: Change “lowered” to “low”.

432-439: I find the conclusions a bit short. Maybe mention a few of your most important results and point out what effects they will have. A clearer punchline is needed.

Are these differences “good” or “bad”? You say that they might explain the low survival in released farmed mallards. Maybe it is good that their survival is low, otherwise would potentially more of them introgress the wild population, leading to negative consequences. But from an animal welfare point of view, a low survival and possibly suffering released individuals is not so positive. Also, the complement activity was higher in farmed mallards, isn’t that a positive trait that could be inferred to the wild population? Could you elaborate on how, and if, something should be done to change the practice of rearing mallards in captivity? Could you put your results in a wider perspective, could they also be useful in other systems than just mallards?

Figure captions

660: I think that Czech Republic should be mentioned somewhere here in the caption.

661-662: Change to: “Sample sizes (n) for each location is shown together with numbers of hatched individuals, in parentheses.”

664-665: The captions should include what’s in figures A-E. After body mass is tarsus length and then relative tarsus length.

In the figure, change “Tarzus” to “Tarsus”.

What does it mean that the relative variables (C-E) are on a negative scale?

669: The captions should include what’s in figures A-E. Please state all variables in the text.

In the figure, change “Tarzus” to “Tarsus”.

674-675: The figures do not say anything about sex. But they do show how H/L ratio and body mass correlate in the three different stages (3-, 9- and 15-days old). Please correct this.

685: According to your material and methods and results, the sample size should be 64.

In the figure, change “halflife” to “half-life”.

Table 1

To follow the same order as in the table and other figures, place relative bill width before relative bill length.

In the table, change “tarzus” to “tarsus”.

Maybe the age categories need to be explained here? What do Age, Age2 and Age3 mean? I don’t think that you explain it in the text either.

Table 4

Add “as” before “a proxy of…”.

The sample size here is not the same as in the text or in figure 6.

6. PLOS authors have the option to publish the peer review history of their article (what does this mean?). If published, this will include your full peer review and any attached files.

Reviewer #1: Yes: Jocelyn Champagnon

Reviewer #2: No

---

## [Author Response · Author response to Decision Letter 0]

31 May 2020

Magdalena Ruiz-Rodriguez

Academic Editor

PLOS ONE

Dear Professor Ruiz-Rodriguez,

herein, we are re-submitting our manuscript PONE-D-20-03121 “Differences in the growth rate and immune strategies of farmed and wild mallard populations” to be further considered for publication in the PLOS ONE. We truly appreciate very constructive review of both Referees and took all of his comments and recommendations into account during the revision of the manuscript. In most cases, we agreed with their criticism and conducted corresponding changes in the manuscript. Detailed list of our responses and text changes is provided bellow. We believe that current manuscript version will be interesting for broad audience of Plos One readers.

On behalf of all authors,

Yours sincerely,

Jana Svobodová (corresponding author) 

Jana Svobodová

Department of Ecology

Czech University of Life Sciences Prague

Kamýcká 129

CZ- 165 00 Prague 

Czech Republic, Europe

Phone: +420 224 383 852

svobodovajana@fzp.czu.cz

Additional requirements

Response: Our manuscript meets PLOS ONE's style requirements.

2. In your Methods section, please provide additional information regarding the permits you obtained for the work. Please ensure you have included the full name of the authority that approved the collection sites access and, if no permits were required, a brief statement explaining why.

Response: The Ethics Committee of the Central Commission for Animal Welfare at the Ministry of Education, Youth and Sports of the Czech Republic approved this research with animals (No. MSMT-31220/2014-5). This information has been added in Methods (l. 163-166).

3. As part of your revision, please complete and submit a copy of the ARRIVE Guidelines checklist, a document that aims to improve experimental reporting and reproducibility of animal studies for purposes of post-publication data analysis and reproducibility: https://www.nc3rs.org.uk/arrive-guidelines. Please include your completed checklist as a Supporting Information file. Note that if your paper is accepted for publication, this checklist will be published as part of your article."

Response: The checklist has been completed and included as a supplement file.

4. In your Methods section, please include a comment about the state of the animals following this research. Were they euthanized or housed for use in further research? If any animals were sacrificed by the authors, please include the method of euthanasia and describe any efforts that were undertaken to reduce animal suffering."

Response: Ducklings from our experiment have not been euthanized. We have given them to several different breeders after termination of our experiment. This statement has been added in the Methods (l. 162-163).

Responses to Reviewers´ comments

PONE-D-20-03121

Title: Differences in the growth rate and immune strategies of farmed and wild mallard populations

Reviewer #1: SUMMARY

This article addresses the question of the low survival in the wild of captive-bred mallards released for hunting. It compares the early duckling’s development (over the first 20 days) of wild and captive-bred mallards in a control captive experiment, focusing on morphological, stress and immune parameters.

While this topic is interesting and the study design appropriate, the article presents some mistakes and the conclusion is not totally supported by the results. I suggest below some revisions to meet the journal's requirements.

Response: We do not agree that our manuscript addresses the question of the low survival in the wild of captive-bred mallards released for hunting. Instead, we compared divergence in phenotype between the two groups (l. 85-98). It is tempting to speculate that these differences may affect fitness under natural conditions. However, we did not intended to quantitatively examine consequences of observed phenotype divergence.

MAJOR ISSUES

First, some claims are not sufficiently justified, in particular the conclusions (lines 432-439)

The results suggest an appropriate response of captive-bred mallards to survive in the wild. With higher maternal investment, captive-bred ducklings had a growth rate, haematopoiesis and immune system that looks rather suitable to survive in a wild environment. Nevertheless, the authors concluded to a divergence with wild mallard that may contribute to the lower survival of captive-bred mallards in a natural environment. Because the results did not totally support this statement, the hypotheses and mechanisms explaining the claim should be detailed.

Response: We rephrased and partly rewrote corresponding parts of the text to make clear distinction between phenomenological findings of our study and their putative conservation implications. 

In addition, even if the results suggest that the populations are genetically divergent, this result has been already shown in other studies (through more robust analysis of population genetics) and by focusing on early development, this study does not contribute with new information to claim that massive restocking could lead to a decrease in fitness of wild populations (last sentence in the abstract and conclusions). Here also, the authors must argue their point through the description of the mechanism, based on their results.

Responses: Following review’s recommendation, last sentences of the abstract and conclusions were rewritten. We would like to recall that the aim of our contribution was not to demonstrate genetic divergence or fitness differences between the two populations. Consequently, we fully agree with the reviewer that our manuscript does not provide any new results on these aspects. Nevertheless, regarding the main aim of our manuscript, the relevance of this particular objection is rather questionable (l. 36-49).

Second, I identified some errors that could be typos but could also reflect more serious concerns on the text-to-figures matching.

First the sample size in the methods: In the text, line 117-123, it is written that 26 and 43 eggs were collected in the wild and in hunting farms respectively, but it contradicts the figure 1 (37 and 64 eggs were collected respectively). From the results, I understood that the numbers presented in the methods refer to the numbers of eggs hatched and not collected.

Response: We are sorry for this inaccuracy. Information presented in this section of original submission regarded numbers of hatched individuals. We corrected it on the new manuscript version (l. 238-243).

Second and more importantly, the results presented in line 264-265 contradicts the figure 3. The slopes in the text are negatives while in the figures 3D and 3E, the slopes of wild mallard subset look clearly positives.

Response: Thank you for pointing out to this mistake. We corrected regression coefficients in current text version (l. 275-278).

Finally, all the tables, figures and supplementary materials present an error on the word "tarsus" written "tarzus"

Response: It has been corrected to tarsus throughout the manuscript (e.g. Tab 1, Fig. 1, Fig. 2).

Third, the method section is not sufficiently detailed on the ethics because it did not describe the fate of the ducklings after 20 days. Were they sacrificed? Journal requires that if anaesthesia, euthanasia, or any kind of animal sacrifice is part of the study, the article should include briefly which substances and/or methods were applied.

Response: We added requested details in the text. The ducklings were not sacrificed. They were given to different breeders after completion of the experiment (l. 162-163).

Fourth, the authors stated that “all relevant data are within the manuscript and its supporting information files”. Nevertheless, only the results of the model are available in the text and the supplementary materials, not the raw data.

Response: According to journal policy, row data should be provided upon manuscript acceptance. We have included S2 Table including all variables that were used in statistical model. In addition basic descriptive summary of morphometric parameters variation between wild vs. farmed group and between age classes is provided as a separate sheet of S1 Table. 

MINOR ISSUES

line 63-65: It is not appropriate to make a numerical comparison between the number of European breeding pairs and the number of juveniles released in the region, as these breeding pairs can potentially give birth to more than 10-12 ducklings each. In addition, the pairs breed in spring while the juveniles are released in summer, primarily at the onset of the hunting season.

Response: The sentence was partly rewritten (l. 60-63). We agree with the reviewer that compassion of these two numbers does not allow to make any general conclusions (which was actually not done in the text). But we believe that presenting [1] number of individuals in wild population along with [2] number of individuals released every year by hunters and breeding facilities is appropriate way how to provide readers insight into the extent of mallard restocking. We were grateful to reviewer for any other metric that would illustrate the situation in more appropriate way and we are willing to modify this section based on his/her specific recommendations. 

By the way, it is not totally true that farmed juveniles are always released during the summer. In the Czech Republic, this take place typically during late May or beginning of June (i.e. shortly after the peak of the breeding season). 

line 125: Please specify whether the collection of eggs in captivity took place from newly laid eggs. If not please detail the age of the eggs as presented for the collection of wild eggs.

Response: The collection of eggs in captivity were made from newly laid eggs. This information has been added in the text (l. 125-127).

line 354-357: This argument needs to be more cautious. The lack of detection of a non-linear asymptotic relationship may be due to a lack of statistical power, the linear model being more parsimonious in terms of number of parameters.

Response: We agree and rephrased the sentence in order to download this statement (l. 375-376). On the other hand, results of these tests were very far from being significant. Consequently, we do not believe that low statistical power could have been the main reason for lack of significance.

OTHER COMMENTS:

line 141: Were the ducklings marked at the age of one day? Please specify.

Response: All ducklings were marked with coloured rings at the age of one day. We specified this information more clearly at l. 143.

line 238: "a significant increase" is not clear because both populations showed it. Authors probably mean that "Body mass[...] showed a significant higher increase in the farmed population [...]".

Response: We agree. It has been replaced by “steeper” (l. 249-250).

line 324: The same problem as above arises and I suggest replacing "increased" with "higher" here.

Response: It has been corrected (l. 339).

Supplementary materials: I found the presentation of the tables without any text rather sketchy. I suggest also to highlight the significant results in the tables. In addition, the predictors probably refer to different models but only the P values are presented. I suggest to include model selection with deviance, etc.

Response: We agree and provide improved version of supplementary material along with current manuscript version to make model selection process more transparent: Specifically, [1] We highlighted predictors included in the final MAM by boldface. [2] Order of deleted variables during the step-wise deletion process is indicated. [3] It is true that “predictor” actually means predictor eliminated from given model. Now we described this more clearly in the text and table headers. [4] Please note that the model selection was based on deviance changes (assuming its χ2 distribution) and these values are presented in the corresponding columns (S2 Tab).

Reviewer #2: Restocking wild populations with captive-bred individuals is common not only to help threatened populations but also to boost huntable populations and increase hunting bags, which is the case in Europe for the mallard. Earlier studies have shown both morphological and genetical differences between these released individuals and their free-living conspecifics. The two groups are also hybridizing resulting in potential negative effects for the wild population. The present study aims to growth-parameters as well as immune functions in both captive-bred and wild individuals in a common-garden experiment. Their results show differences in both morphology, growth-rates and immune functions and are explained by both genetical and environmental factors.

It is an interesting and important study that contribute to the understanding of the effects of large-scale releases of captive-bred individuals. Although it deals with questions that, at least partly, already have been studied in other articles, I believe that it adds to research area with a smart and simple experimental set-up that focus on the important period of a captive-bred ducklings’ life when it is in the hands of humans.

The manuscript is well-written and relatively easy to take in. The authors should however try to elaborate on their findings, specially about the immune functions and also put them in a wider perspective.

Comments below reference to the line numbers in the manuscript.

Response: We are grateful for all your comments on the manuscript and believe that they helped us to improve significantly quality of our work. We took all your concern into account and conducted corresponding changes in the manuscript text. Following your recommendations, we also attempted to provide more elaborate version of discussion and conclusions in current manuscript version. 

Abstract

27: Change “anatomical” to “morphological”.

Response: It has been changed (l. 28).

29: Change “in free-living populations and those from breeding facilities” to “between free-living populations and individuals from breeding facilities”.

Response: It has been changed (l. 29).

34: Maybe “greater” is more correct than “higher”?

Response: This part has been rewritten according suggestion of Reviewer #1.

38: If the result is negative maybe “due to” is more appropriate.

Response This part has been rewritten according suggestion of Reviewer #1.

Introduction

61: Delete “on” and “of” (before “duck”).

Response: It has been deleted (l. 59).

62: Change “stocking” to “restocking”.

Response: It has been changed (l. 60).

63: Maybe worth mentioning somewhere, the extent of releases in Czech Republic?

Response: The extent of mallard releases in the Czech Republic has been mentioned (l. 115-117).

63: In the manuscript you use several different terms for these mallards: Captive-bred, captive-reared, farmed etc. And there exists many more in the different articles you cite. Maybe worth mentioning that several different terms are used (sometimes with different meanings) and that you in this manuscript use the following… Stick to one or two and be consistent. It could also be good to state what you mean with the terms you use.

Response: We agree with this concern. Consequently, we unified the terminology in current manuscript version. “Captive-bred” or “captive-reared” individuals are now consistently called “farmed” throughout the text (l. 41, 67, 72, etc.). 

70: Add reference.

Response: References have been added (l. 68).

79: Full stop after references. Change next sentence to: “However, the phenotypic changes observed in such cases will not necessarily have been caused by restocking alone, but also other factors, such as habitat and climate change, may be of comparable importance [30].”

Response: It has been changed (l. 77-80).

84: Delete “there”.

Response: It has been deleted (l. 82).

88: Change to: “(up to an age of 20 days)”.

Response: It has been corrected (l. 86).

90: Delete extra space after references.

Response: It has been corrected (l. 88).

102-105: I think you need to be clearer here about what you do and what you are testing. I am not sure what prediction you are testing here. When you rear them both in captivity and predict that wild eggs hatched and reared in captivity will have a lower growth and haematopoiesis compared to captive-reared, you at the same time predict that this is genetically determined, right?

Response: We have modified this sentence to make our predictions more clear (l. 101-104).

Methods

116-117: You collected 37 wild eggs from four localities. If you only want to include the hatched eggs, you should also mention that they are only from three localities. But I think that is better to mention in the results.

Response: We agree and information on sample size has been moved to “Results” section (l. 238-243). 

121: You mean less than (<) four days?

Response: Yeas. It has been corrected (l. 122).

122-123: You collected 64 farmed eggs.

Response: The information on sample size has been moved to “Results” section (l. 238-243).

141: Delete extra space after references.

Response: It has been deleted (l. 141).

142: At the end of the sentence, add: “in each cage”.

Response: It has been added (l. 145).

150: Rephrase sentence using passive form and avoid pronouns.

Response: The sentence has been rephrased to passive form (l. 152-155).

152-153: With an accuracy of 0.01mm?

Response: The accuracy and type of digital calliper have been added (l. 156).

152: How was the bills measured? Over the nostrils? Describe and possibly give a reference.

Response: We specified these details at l. 152-155. Bill length was measured from the tip to the feathering and bill width in the maximal width of it’s distal part.

153-154: Why not sample them the same days as you measured them?

Response: The reason was that leucocytes profile can be changed rapidly by short-term stress (e.g. by handling, Cīrule et al. 2012). Therefore drop of blood was taken from the jugular vein on the 3rd, 9th and 15th day (i.e. one day interval between morphometric measurements and blood sampling) and used to prepare a blood smear for haematological analysis (l 156-161).

153-155: Type of syringe and size of needle?

Response: It has been added (l. 160).

177: Rephrase sentence and avoid using pronouns.

Response: The sentence has been rephrased to passive form (l. 184).

188-189: Have you considered using the residuals of a body mass and tarsus length/bill measurements regression? This is common when using body condition indices. See e.g:

Green, A.J. (2001). Mass/length residuals: measures of body condition or generators of spurious results? Ecology, 82, 1473-1483.

Jakob, E.M., Marshall, S.D. & Uetz, G.W. (1996). Estimating Fitness: A Comparison of Body Condition Indices. Oikos, 77, 61-67.

Schulte-Hostedde, A.I., Zinner, B., Millar, J.S. & Hickling, G.J. (2005). Restitution of mass–size residuals: validating body condition indices. Ecology, 86, 155-163.

In any way, wouldn’t body mass divided by size be a more appropriate way to calculate the variable?

Response: We appreciate this insightful comment. In initial steps of our analyses, we considered these or similar approaches (specifically Peig and Green 2009; Oikos). Nevertheless, due to specific structure of our data (detailed bellow) we decided stick to tarsus length standardized by body mass. We believe that our approach is more transparent and that implementation of commonly used body condition measures would be problematic and could lead to spurious results. Specifically, using residuals for the purpose of body condition analyses make sense in homogenous population sample. However, this is not the case of our dataset. We were working with two populations of contrasting developmental trajectories. Moreover, the samples size for the two populations was not the same. Common regression (necessary for residual-based body condition analyses) would have poor fit, i.e. would lead “in between” these two populations. Consequently, normality of residual (basic assumption on regression models) cannot be meet. Moreover, the regression slope could by biased due to uneven sample size for the two groups. Finally, the fact that ducklings were measured during 20 days of their rapid growth precludes straightforward and transparent calculation of body condition. Consequently we believe that tarsus length – body mass ratio provides simple and the most transparent way, how to address this task. In the case that this argumentation will not satisfy the reviewer, we are willing to completely delete these analyses from the manuscript as we do not find them indispensable. 

201: You could only measure to nearest 0.01mm, so four decimals feels a bit too much.

Response: It has been corrected to two decimals (l. 208).

209: Change to: “specified as random effects…”

Response: The sentence has been corrected (l. 216).

223-228: I assume you use a 0.05 significance level, maybe best to state that clearly.

Response: The 0.05 significance level has been added (l. 229).

229: I am not sure on what level raw data should be uploaded/available for others but a supplementary file with all measurements and test values could be of interest for other researchers.

Response: Based on our understanding of journal policy, raw data should be available upon manuscript acceptance. Consequently, we provide raw data associated with this study as a part of supplementary materials (S1 Tab.).

Results

231: How many males and females in each group? Was the sex ratio equal in both groups?

Response: We provided information on sex ratio in each group. Using chi-squared test, we found no difference between farmed and wild group (l. 243-245). 

238-241: You mean a significant higher body mass and tarsus length in farmed compared to wild? When you write increase it sounds like you are talking about growth rate. I guess you tested the differences with a t-test or similar? Please give p-values and test-values.

Response: The sentence was rephrased (l. 249-253). Actually, there are two take home massages: 1) Body mass and tarsus exhibited higher growth rates (as shown in S1Tab. and Fig. 1) and 2) the final value of both these parameters was higher in 20-days-old farmed ducklings (tested by t-tests). 

241-243: You give the measurements after 20 days and conclude that relative tarsus length was higher in the wild after 20 days. And then state that it remained higher throughout the experiment. But the experiment was over after 20 days? So, maybe rephrase by writing that it was higher directly after hatching and remained so after 20 days. Or simply change “remained” to “was”

Response: “Remained” has been changed to “was” (l. 255).

243-244: Mild effect? If I look at the correct values, the p-value is 0.59. I would not call that a mild effect. Or should it be 0.059? Or do you mean in age2 but not in the other age-groups? Age2 is significant so no need to call it a mild effect.

Response: We found a significant interaction between sex and age2 suggesting a significant sex effect on increase of tarsus length (Tab. 1). The sentence has been rewritten in this respect (l. 255-257). 

244: Change “then” to “than”. It is a bit confusing when you sometimes use structural body size and sometimes relative tarsus length.

Response: The sentence has been rewritten (l. 255-257). Now we use “structural body size “ throughout the text (e.g. l. 253, 352).

245-255: Sounds like discussion.

Response: The sentence has been deleted (l. 257).

259-260: The parenthesis is quite important because I don’t know if the actual bill length or width differ between wild and farmed in your study. There is no data that shows the actual measurements of the bills. I think that should be included, at least in the supplemental information.

Response: Following the recommendation of the reviewer parentheses were removed to stress the fact that relative morphophonemic indexes were used (l. 270-271). We prefer using bill measurements standardized by body size in statistical models to avoid reporting mutually correlated results (e.g. individuals of bigger body size have longer bills and wider bill etc.). Nevertheless, we agree that basic overview of morphometric parameters should be provided. Therefore, we present raw data associated with this study and basic descriptive statistics as a part of S1Table.

261-263: Again, did it differ? Or do you mean then relative sizes?

Response: “Remained” has been changed to “was” (l. 273).

268-269: Discussion.

Response: The sentence has been deleted (l. 279).

274: Change “association” to “correlation”.

Response: It has been changed (l. 284).

282: Change ”an increased” to “a higher”.

Response: It has been changed (l. 293).

294: Delete “slight”. There was a significant difference according to your test.

Response: It has been deleted (l. 309).

Discussion

337-338: I would argue that wild mallards have a lower weight in relation to their tars, i.e. they have a lower body condition, therefore, I would divide weight with the size variable (or preferably calculate body condition by using the residuals of a body mass and tarsus length regression, as proposed above).

Response: We agree with this argumentation and we explicitly discuss body condition in this text section. We also restructured this paragraph to stress this aspect. However, (as already explained above,) we prefer focusing our analyses on relative body size than on any alternative body condition proxies. 

338-343: Well, the tarsus is still shorter in wild mallards. What you write in 342-343 is probably correct and would therefor also mean that body mass is probably more affected than tarsus length under natural conditions with a limitation of food.

Response: We agree with the reviewer. This statement has been added in the discussion (l. 359-361).

340: “Structural tarsus length” is not a term you have used before. You mean relative tarsus length i.e. structural body size?

Response: This part has been rewritten (l. 353-358). Nevertheless we used “structural body size” throughout the text (please see our explanation above).

367: Is this correct? Or do you again mean relative sizes? The actual sizes don’t have to be longer just because the relative sizes are. Again, it would be interesting to see the numbers on the actual sizes. You also state that wild had wider bills than farmed but further down (line 376) you are talking about a decrease. Because you have not compared mallards from different time periods, I think you should be cautious to use phrases like increased or decreased when you just mean that your two groups are different to each other. You can talk about increased if you have a baseline or compare the same variable over time. But here you often use it when one group have a higher value than the other.

Response: Yeas, we mean relative length and width bill. The sentence was reworded and the ambiguity regarding “decrease” is no more present in the current manuscript version. In addition, we feel that the usage of relative bill measures is more appropriate that absolute values from several reasons. Nevertheless, to make our analyses more transparent, raw data were provided as a part of supplementary materials of the current manuscript version (l. 385-390). 

375-377: No, that is not correct. According to Söderquist et al. 2014, historical wild mallards had the narrowest bills while farmed had the widest, placing contemporary wild mallards in the middle. So, wild mallards showed an increase in bill width since the start of releases of farmed mallards.

Response: We are sorry for this misinterpretation. The sentence has been rewritten (l. 397-400).

379: Change “lamella” to “lamellar”.

Response: It has been corrected (l. 394).

385: Change “association” to “correlation”.

Response: It has been changed (l. 405).

390: delete extra space before parenthesis.

Response: It has been deleted (l. 410).

400: Chane “than” to “compared to”.

Response: It has been changed (l. 420).

403-406: I find this a bit confusing. There was a positive correlation in 3-day-old, the higher weight, the higher production of erythrocytes. But the correlation changed to negative in 9- and 15-day-old. In 9- and 15-day-old the light individuals have the highest production of erythrocytes. And it is these individuals that you mean compensate for low production during earlier stages? Because the light ones in 9- and 15-day-old were probably also light in 3-day-old, i.e. they had a low production at that stage? Consider rewriting this part to make it more clear which parts of the graphs you are talking about.

Response: Thank you for pointing out to this ambiguity. Ducklings typically retained relatively low (or high) body mass during the whole experiment. Consequently, the change of the sign of the regression coefficient for body mass vs. immature erythrocytes from positive to negative suggests that low haematopoiesis in 3-day-old individuals of low body mass was later compensated by higher haematopoiesis. To support this prediction, we added correlation analysis that compares immature erythrocytes proportions between different age cohorts (l. 302-305). In addition, we rephrased these sentences to avoid further confusion (l. 419-431). 

415: Change “decreased” to “lower”.

Response: It has been changed to “lower” (l. 440).

418: Change “lowered” to “lower”.

Response: It has been changed (l. 443).

435: Change “lowered” to “low”.

Response: The conclusion has been rewritten (l. 456-480).

432-439: I find the conclusions a bit short. Maybe mention a few of your most important results and point out what effects they will have. A clearer punchline is needed.

Are these differences “good” or “bad”? You say that they might explain the low survival in released farmed mallards. Maybe it is good that their survival is low, otherwise would potentially more of them introgress the wild population, leading to negative consequences. But from an animal welfare point of view, a low survival and possibly suffering released individuals is not so positive. Also, the complement activity was higher in farmed mallards, isn’t that a positive trait that could be inferred to the wild population? Could you elaborate on how, and if, something should be done to change the practice of rearing mallards in captivity? Could you put your results in a wider perspective, could they also be useful in other systems than just mallards?

Response: We agree. This text section has been rewritten (l. 456-480).

Figure captions

660: I think that Czech Republic should be mentioned somewhere here in the caption.

Response: It has been added (l. 708).

661-662: Change to: “Sample sizes (n) for each location is shown together with numbers of hatched individuals, in parentheses.”

Response: It has been rewritten according suggestion of the reviewer (l. 710-711).

664-665: The captions should include what’s in figures A-E. After body mass is tarsus length and then relative tarsus length.

In the figure, change “Tarzus” to “Tarsus”.

What does it mean that the relative variables (C-E) are on a negative scale?

Response: Figures A-E have been described (l. 713-715). “Tarzus” has been corrected to “tarsus” (Fig. 2). We are sorry for the confusion regarding negative scale in some of these plots. This is a consequence of logarithmic transformation that was used for most morphometric parameters (as mentioned at l. 202-203 of the current submission). Now we specify the scale of response variables and predictors more clearly in the axes titles of corresponding plots.

669: The captions should include what’s in figures A-E. Please state all variables in the text.

In the figure, change “Tarzus” to “Tarsus”.

Response: It has been corrected (l. 720-721, Fig. 3).

674-675: The figures do not say anything about sex. But they do show how H/L ratio and body mass correlate in the three different stages (3-, 9- and 15-days old). Please correct this.

Response: It has been corrected (l. 725-726).

685: According to your material and methods and results, the sample size should be 64.

In the figure, change “halflife” to “half-life”.

Response: Axis title was changed. In the case of complement sample size was 61 because we did not succeed to collect sufficient plasma volume for all individuals. Consequently, complement activity was measured for 38 farmed and 23 wild ducklings only (l. 184-185). The sample size has been corrected (l. 736). 

Table 1

To follow the same order as in the table and other figures, place relative bill width before relative bill length.

In the table, change “tarzus” to “tarsus”.

Maybe the age categories need to be explained here? What do Age, Age2 and Age3 mean? I don’t think that you explain it in the text either.

Response: Tab. 1 has been corrected according suggestion of the reviewer. 

Table 4

Add “as” before “a proxy of…”.

The sample size here is not the same as in the text or in figure 6.

Response: It has been corrected (Tab. 4 caption). The sample size is lower because a sufficient plasma volume were not possible to collect for all individuals. Therefore complement activity was measured for 38 farmed and 23 wild ducklings only (l. 184-185).

References

Cīrule D., Krama T., Vrublevska J., Rantala M. J. & Krams I. 2012: A rapid effect of handling on counts of white blood cells in a wintering passerine bird: A more practical measure of stress? Journal of Ornithology 153: 161-166.

Peig J. & Green A. J. 2009: New perspectives for estimating body condition from mass/length data: The scaled mass index as an alternative method. Oikos 118: 1883-1891.

---

## [Decision Letter · Decision Letter 1]

16 Jun 2020

PONE-D-20-03121R1

Differences in the growth rate and immune strategies of farmed and wild mallard populations

PLOS ONE

Dear Dr. Svobodová,

Thank you for submitting your manuscript to PLOS ONE. After careful consideration, we feel that the manuscript needs some minor changes before publication. Below you can find the suggestions made by the Reviewer and me. Therefore, we invite you to submit a revised version of the manuscript that addresses the points raised during the review process.

We look forward to receiving your revised manuscript.

Kind regards,

Magdalena Ruiz-Rodriguez

Academic Editor

PLOS ONE

Additional Editor Comments

This new version has been really improved with the changes performed according to reviewers suggestions.

In addition to the comments to this new version made by the Reviewer 2, I would like to add the following:

L36: change “demonstrated” to “demonstrate”

L373: change “worlds” to “words”

L420: change compare to “compared”

Reviewer's Responses to Questions

**Comments to the Author**

1. If the authors have adequately addressed your comments raised in a previous round of review and you feel that this manuscript is now acceptable for publication, you may indicate that here to bypass the “Comments to the Author” section, enter your conflict of interest statement in the “Confidential to Editor” section, and submit your "Accept" recommendation.

Reviewer #2: (No Response)

2. Is the manuscript technically sound, and do the data support the conclusions?

Reviewer #2: Yes

3. Has the statistical analysis been performed appropriately and rigorously? 

Reviewer #2: I Don't Know

4. Have the authors made all data underlying the findings in their manuscript fully available?

Reviewer #2: Yes

5. Is the manuscript presented in an intelligible fashion and written in standard English?

Reviewer #2: Yes

6. Review Comments to the Author

Reviewer #2: I am satisfied with the answers to my questions and comments on the first draft of the manuscript. I feel like the authors have listened to the comments and improved the manuscript. I still have some minor suggestions for the authors to deal with.

Abstract

L34: Change “parameters and a higher growth rates and higher complement activity” to “parameters, a higher growth rate, and higher complement activity”.

L37: Change “population” to “populations”

L38: Change “prose” to “argue”.

L39: Add “the” before “breeding population”.

Introduction

L41: delete “as” before “hand-reared”.

L59: add “of” before “the most widespread”

Material & methods

L118: Maybe there was some misunderstanding. I still think that you need to give the number of collected eggs here in the m&m. If you will present that not all eggs hatched successfully, that probably belongs in the results. The most important thing is that the numbers add up.

L157: Check spelling of “leukocyte” in whole document, be consistent.

L208: Change “mm3” to “mm3”.

Results

L249-253: This sentence about the slope of body mass and tarsus length is not connected to the t-test in the following sentence, right? The t-test is just for the final measurements of body mass and tarsus? Maybe start with that the final measurements were significantly different and give the t-test and then continue with that also the slopes were different. Or, give test-values for the first statement that the slopes were different.

Also suggested change of sentence, from ”the average body mass was 321.6 ± 7.6 g (mean ± S.E) and tarsus length 46.4 ± 0.3 mm in 20-day-old farmed duckling, compared with 238.9 ± 11.3 g and 42.2 ± 0.5 mm (t-test: p < 0.0001 in both cases), respectively, in wild ducklings of the same age” to: ”body mass and tarsus length differed significantly between farmed and wild 20-day-old ducklings (mean ± S.E: 321.6 ± 7.6 g, 46.4 ± 0.3 mm, and 238.9 ± 11.3 g, 42.2 ± 0.5 mm, respectively; t-test: p < 0.0001 in both cases).”

L300-306: Be consistent when writing their age (3-day-old, 9-days old, 15-day- old, 3-days-old etc). Check whole manuscript.

Discussion

L354: Change “non-sceletal” to “non-skeletal”.

L359: Change “means” to “mean”.

L360: Change “with” to “by”.

7. PLOS authors have the option to publish the peer review history of their article (what does this mean?). If published, this will include your full peer review and any attached files.

Reviewer #2: Yes: Pär Söderquist

---

## [Author Response · Author response to Decision Letter 1]

3 Jul 2020

Magdalena Ruiz-Rodriguez

Academic Editor

PLOS ONE

Dear Professor Ruiz-Rodriguez,

herein, we are re-submitting our manuscript PONE-D-20-03121 “Differences in the growth rate and immune strategies of farmed and wild mallard populations” to be further considered for publication in the PLOS ONE. We have conducted changes in the manuscript according to your comments and suggestions of the Reviewer 2. Detailed list of our responses and text changes are provided in the Cover Letter.

On behalf of all authors,

Yours sincerely,

Jana Svobodová (corresponding author)

---

## [Editor Report · Decision Letter 2]

10 Jul 2020

Differences in the growth rate and immune strategies of farmed and wild mallard populations

PONE-D-20-03121R2

Dear Dr. Svobodová,

We’re pleased to inform you that your manuscript has been judged scientifically suitable for publication and will be formally accepted for publication once it meets all outstanding technical requirements.

Kind regards,

Magdalena Ruiz-Rodriguez

Academic Editor

PLOS ONE

---

## [Editor Report · Acceptance letter]

10 Aug 2020

PONE-D-20-03121R2 

Differences in the growth rate and immune strategies of farmed and wild mallard populations 

Dear Dr. Svobodová:

I'm pleased to inform you that your manuscript has been deemed suitable for publication in PLOS ONE. Congratulations! Your manuscript is now with our production department. 

Kind regards, 

on behalf of

Dr. Magdalena Ruiz-Rodriguez 

Academic Editor

PLOS ONE